# GC-MS Analysis, Antibacterial, and Anticancer Activities of *Hibiscus sabdariffa* L. Methanolic Extract: In Vitro and In Silico Studies

**DOI:** 10.3390/microorganisms11061601

**Published:** 2023-06-16

**Authors:** Amira E. Sehim, Basma H. Amin, Mohammed Yosri, Hanaa M. Salama, Dalal Hussien Alkhalifah, Maha Abdullah Alwaili, Rasha Y. Abd Elghaffar

**Affiliations:** 1Botany and Microbiology Department, Faculty of Science, Benha University, Benha 13518, Egypt; amira.alsayed@fsc.bu.edu.eg (A.E.S.); rasha.mohamed@fsc.bu.edu.eg (R.Y.A.E.); 2The Regional Center for Mycology and Biotechnology, Al-Azhar University, Cairo 11787, Egypt; mohammedafifi.18@azhar.edu.eg; 3Department of Chemistry, Faculty of Science, Port Said University, Port Said 42521, Egypt; hana_negm2020@yahoo.com; 4Department of Biology, Collage of Science, Princess Nourah Bint Abdulrahman University, P.O. Box 84428, Riyadh 11671, Saudi Arabia; maalwaele@pnu.edu.sa

**Keywords:** medicinal plants, antibacterial, MDR, anticancer, GC-MC, scanning electron microscope, docking study

## Abstract

The emergence of bacteria that are resistant to several antibiotics has represented a serious hazard to human health globally. Bioactive metabolites from medicinal plants have a wide spectrum of therapeutic possibilities against resistant bacteria. Therefore, this study was performed to investigate the antibacterial efficacy of various extracts of three medicinal plants as *Salvia officinalis* L., *Ziziphus spina-christi* L., and *Hibiscus sabdariffa* L. against pathogenic Gram-negative *Enterobacter cloacae* (ATCC13047), *Pseudomonas aeruginosa* (RCMB008001), *Escherichia coli* (RCMB004001), and Gram-positive *Staphylococcus aureus* (ATCC 25923), bacteria using the agar-well diffusion method. Results revealed that, out of the three examined plant extracts, the methanol extract of *H. sabdariffa* L. was the most effective against all tested bacteria. The highest growth inhibition (39.6 ± 0.20 mm) was recorded against *E. coli*. Additionally, the minimum inhibitory concentration (MIC) and the minimum bactericidal concentration (MBC) of the methanol extract of *H. sabdariffa* were detected in the case of all tested bacteria. Moreover, an antibiotic susceptibility test revealed that all tested bacteria showed multidrug resistance (MDR). While 50% of tested bacteria were sensitive and 50% were intermediately sensitive to piperacillin/tazobactam (TZP) based on the inhibition zone but still less than the extract. Synergistic assay demonstrated the promising role of using a combination of *H. sabdariffa* L. and (TZP) against tested bacteria. A surface investigation using a scanning electron microscope of the *E. coli* treated with TZP, extract, or a combination of the two revealed extremely considerable bacterial cell death. In addition, *H. sabdariffa* L. has a promising anticancer role versus Caco-2 cells with IC_50_ of 17.51 ± 0.07 µg/mL and minimal cytotoxicity upon testing versus Vero cells with CC_50_ of 165.24 ± 0.89 µg/mL. Flow cytometric analysis confirmed that *H. sabdariffa* extract significantly increased the apoptotic rate of Caco-2-treated cells compared to the untreated group. Furthermore, GC-MS analysis confirmed the existence of various bioactive components in the methanol *hibiscus* extract. Utilizing molecular docking with the MOE-Dock tool, binding interactions between n-Hexadecanoic acid, hexadecanoic acid-methyl ester, and oleic acid, 3-hydroxypropyl ester were evaluated against the target crystal structures of *E. coli* (MenB) (PDB ID:3T88) and the structure of cyclophilin of a colon cancer cell line (PDB ID: 2HQ6). The observed results provide insight into how molecular modeling methods might inhibit the tested substances, which may have applications in the treatment of *E. coli* and colon cancer. Thus, *H. sabdariffa* methanol extract is a promising candidate to be further investigated for developing alternative natural therapies for infection treatment.

## 1. Introduction

Global morbidity and mortality rates from infectious diseases have significantly increased, turning them into a critical health issue. In the past decade, several antibiotics have been used to fight these infections [1]. As a result of the misuse or overuse of antibiotics, multidrug-resistant (MDR) bacteria have emerged that are currently rapidly spreading over the globe and represents a threat to global public health [2,3]. Generally, MDR bacteria are resistant to at least one antimicrobial agent from three or more antimicrobial classes by in vitro susceptibility assay [4,5]. Multidrug-resistant (MDR) bacteria, also reported as “ESKAPE” pathogens, including *Enterococcus faecium*, *Staphylococcus aureus*, *Klebsiella pneumoniae*, *Acinetobacter baumanii*, *Pseudomonas aeruginosa*, and *Enterobacter* species, are the main cause of infection and effectively “escape” the effects of antibacterial drugs [6]. Due to the side effects of antibiotics and microbial resistance, its necessary to use alternative plant-based antibiotics to overcome the resistant strains [7].

Human infections are traditionally treated with medicinal plants [8]. Natural compounds derived from plants are the main resource of antimicrobial agents being natural, inexpensive, safe to the host, and active at low concentrations [9,10]. Based on estimates from the World Health Organization (WHO), 80% of people worldwide use extracts of plants or their active ingredients in traditional medicines [11]. Several investigations have emphasized the effective role of medicinal plants in controlling infectious diseases as they have bioactive components, such as flavonoids, phenolics, alkaloids, terpenoids, tannins, essential oils, lectin, polypeptides, and polyacetylenes [12,13,14].

*Salvia officinalis* (Sage) is a member of the Lamiaceae family. Middle Eastern and Mediterranean regions are its native regions. *S. officinalis* has been applied to treat seizures, osteoarthritis, diarrhea, and hyperglycemia in herbal medicine. Recently, it has reportedly been found to have antibacterial and antioxidant effects [15,16,17].

*Ziziphus spina-christi* (*Sedra*) is an evergreen tree or plant with native distribution to North Africa as well as the south and west areas in Asia [18]. It was utilized in Saudi Arabian traditional medicine to cure infectious disease disorders, such as ringworm, palpitations, hypertension, sleeplessness, and diabetes [19]. Previous investigations in modern medicine have demonstrated this plant’s pharmacological capabilities, which include an antimicrobial, antioxidant, anti-diabetic, and anti-cancer impact [20]. *Zizyphus spina-christi* leaves and fruit extract demonstrated antimicrobial or antibacterial activity versus *S. aureus*, *C. albicans*, *B. subtilis*, and *E. coli* [21].

*Hibiscus sabdariffa* (Roselle) is a multiuse medicinal plant from the Malvaceae family. It is an annual tropical short shrub and is spread in many tropical and sub-tropical regions in the world [22]. It has antimicrobial, antioxidant, hypotensive, hypocholesterolemic, immune-modulated, hepatoprotective, renoprotective, diuretic, anti-obesity, antiurolithic, antidiabetic, and anticancer traits without any significant genotoxic effects [23]. Previous work showed the antibacterial activity of *H. sabdariffa* versus *S. aureus*, *S. epidermidis*, *S. enterica*, *K. pneumonia*, *Ps. aeruginosa*, *E. coli*, *P. vulgaris,* and *B. cereus* [24]. Based on the importance of medicinal plants in controlling infectious disease, the current work aimed to compare the antibacterial capability of *S. officinalis*, *Z. spina-christi*, and *H. sabdariffa*, illustrating the synergistic effect of the most promising extract in coordination with various antibiotics versus tested bacteria as well as the anticancer impact of the extract and reporting its role in the apoptotic process.

## 2. Materials and Methods

### 2.1. *Plant Materials and Extraction*

Two plants were bought in dried form from an Egyptian local market: *S. officinalis* L. leaves, often known as sage, and *H. sabdariffa* L. flowers, also known as roselle. Additionally, the professional plant-qualified worker at the Botany and Microbiology Department, Faculty of Science, Benha University confirmed *Z. spina-christi* L. leaves that had been taken from Monufia 30.52° N 30.99° E in Egypt (Voucher No. ID: 0045).

Sage, ziziphus, and rosella powder samples weighing 10 g each were used in the experiment. For water extraction, the plant materials were heated at 90 °C for 30 min in 90 mL of distilled water and then placed overnight at 37 °C in a shaking incubator at 150 rpm. Similar to this, 10 g of each examined plant material powder was combined with ethanol and methanol (99%) separately, before incubation at 37 °C and 150 rpm for an entire night. The resultant liquid extracts were then passed through a Whatman No. 1 filter to separate them from the solid residue and then were evaporated to room temperature [25,26,27]. For testing their antibacterial activity, aqueous extracts were dissolved in distilled water, while methanol and ethanol extracts were diluted in 10% dimethyl sulfoxide (DMSO) to achieve final concentrations of 10 mg/mL.

### 2.2. Bacterial Strains

Pathogenic bacterial strains were provided by AL-Azhar University, Regional Center for Mycology and Biotechnology (RCMB) culture collection unit. The bacteria used in this study were *E. cloacae* (ATCC13047), *S. aureus* (ATCC 25923), *P. aeruginosa* (RCMB008001), and *E. coli* (RCMB004001).

### 2.3. Antibacterial Assay

The tested bacteria were grown in nutrient broth and cultured overnight at 37 °C to achieve the turbidity of 0.5 McFarland standards giving 1.5 × 10^8^ CFU/mL. Antibacterial activity of plant extracts was performed using the agar-well diffusion method [28,29]. Mueller-Hinton agar plates were cultured with bacterial suspensions. Wells (6 mm) were drilled into the inoculated media using a sterile cork- borer. A total of 100 µL of plant extracts (aqueous, methanol, and ethanol) were poured separately into each well. Then, the plates were kept for 30 min in the refrigerator for better diffusion of the plant extracts into the agar. Penicillin G (P, 10 mg) and DMSO (10%) were used as positive and negative controls. The plates were incubated at 37 °C for 24 h. After the incubation period, the zone of inhibition was measured to determine the antibacterial capability.

### 2.4. Determination of Minimum Inhibitory Concentration (MIC) and Minimum Bactericidal Concentration (MBC)

For MIC detection: The plant extract that displayed the highest bacterial growth inhibition using an agar-well diffusion technique was further screened by micro-dilution technique [30,31]. Using broth as diluent, two-fold serial dilutions of methanol extract were prepared from the stock solution to achieve concentrations ranging from (1000 to 1.9 µg/mL). Finally, a volume of 10 μL was taken from a bacterial suspension (10^5^ CFU/mL) and then added to each well. To ensure the sterility and purity of the medium, wells containing an un-inoculated medium with and without extract were used as a control. In order to ensure that the organism could grow in the medium, a third control well was used that contained inoculated medium but no extract. The turbidity was tested as a measure of microbial growth after being incubated for 24 h at 37 °C. The lowest concentration of plant extract dilution, which inhibited any visible growth of the tested bacteria, is considered a MIC value.

For MBC detection: On Mueller-Hinton agar plates, streaks from the plant extract’s lowest concentrations that had no discernible growth were spread out. Then, the plates were incubated at 37 degrees Celsius for 24 h before being checked for bacterial growth in accordance with a plant extract concentration. MBC was found to be the lowest concentration of plant extract on the newly inoculated agar plates that did not show any bacterial growth. Two types of activities were demonstrated by plant extract: a bacteriostatic (MBC/MIC ≥ 4) and bactericidal (MBC/MIC < 4) [32].

### 2.5. Antibiotic Sensitivity Assay

The following antibiotics were tested for antibiotic susceptibility against the test bacteria via the disc diffusion method on Mueller-Hinton agar: vancomycin (VA, 30 mg), meropenem (MEM, 10 mg), gentamicin (CN, 10 mg), erythromycin (E, 15 mg), amikacin (AK, 30 mg), ciprofloxacin (Cip, 5 mg), tetracycline (TE, 30 mg), amoxicillin/clavulonic acid (AMC, 30 mg), piperacillin/tazobactam (TZP, 110 mg), doxycycline (DO, 30 mg), ceftazidime (CAZ, 30 mg), and cefotaxime (CTX, 30 mg). Based on the interpretation criteria defined by the Clinical and Laboratory Standards Institute, the used bacteria were categorized as susceptible (S), moderately susceptible (I), and resistant (R) [33].

### 2.6. Synergistic Assay

According to the antibiotic sensitivity test results, the most effective antimicrobial disc (TZP) was saturated with 25 μL of the methanol extract, allowed to dry, and carefully applied on the surface of inoculated MHA by pressing slightly. The inhibition zones displayed by the plant extract in combination with the antibiotic on the plates were measured after a 24 h incubation period at 37 °C. If the inhibition zone of combination treatment was higher than the zone of plant extract plus the zone of the corresponding antibiotic, this was considered synergism; if it was equal to the inhibition zone of plant extract plus the zone of the corresponding antibiotic, this was taken as additive; and if it was less than the zone of plant extract plus the zone of the corresponding antibiotic, this was regarded as antagonistic [34].

### 2.7. Scanning Electron Microscopy

The morphological changes in untreated and treated *E. coli* with (TZP), *H. sabdariffa* L., methanolic extract, and their combinations as separate treatments were examined using a scanning electron microscope. After being covered with gold and dried in an ethyl alcohol series, preserved specimens were inspected under the microscope (JOEL, Tokyo, Japan) [35,36].

### 2.8. Anticancer and Cytotoxicity Assay

*H. sabdariffa* L. methanol extract was tested for cytotoxic effects on Vero (African green monkey cells) and Caco-2 (colorectal cancer cells). Cells were left to attach for 24 h until confluence, after which they were supplied with the plant extract at concentrations ranging from 500 to 15.63 µg/mL and incubated at 37 °C for 24 h. The fresh medium was then added, and after 4 h at 37 °C, 100 µL of MTT solution (5 mg/mL) was applied. Using a microplate reader, absorbance was found at 570 nm [37,38].

### 2.9. Flow Cytometric Analysis

Both untreated and treated Caco-2 cells with extract were used for this assay. Trypsin in 0.25% pancreatin was used to split the Caco-2 cells, and phosphate-buffered saline was used to wash them. An Annexin V-FITC and propidium iodide staining kit (B.D. Bioscience, San Jose, CA, USA) was used to determine the mortality rate. Cells were floated in a buffer containing Annexin V-FITC and/or P.I. stock solution, and they were kept at room temperature for ten minutes. Flow cytometry was utilized for analysis [39,40].

### 2.10. Gas Chromatography-Mass Spectrometry (GC-MS) Analysis

A direct capillary column TG-%MS (30 m × 0.25 mm × 0.25 m film thickness) was used with a trace GC1310-Isq mass spectrometer (Thermo Scientific, Austin, TX, USA) to analyze the chemical composition of the methanol extract of *H. sabdariffa* L. The temperature of the column oven was first maintained at 50 °C before being raised by 5 °C/min to 230 °C and held for 2 min. A total of 30 °C /min was added to the final temperature of 290 °C, which was then held for two minutes. Helium was employed as the carrier gas, with a constant flow rate of 1 mL/min, and temperatures of the injector and MS transfer line were maintained at 250 and 260 °C, respectively. Using the split mode of the GC and the Autosampler AS1300, a diluted sample of 1 μL was automatically injected. Mass spectra were gathered in full scan mode from m/z 40 to 1000 at 70 eV ionization voltage. The temperature of the ion source was fixed at 200 °C. By comparing the components’ retention time and mass spectra to those in the WILEY 09 and NIST 11 mass spectral databases, the components were determined [41].

### 2.11. Molecular Docking

In order to investigate the interaction between the ligands and crystal structure of *E. coli* MenB in complex with substrate analogue, OSB-NCoA (PDB ID:3T88) and the structure of the cyclophilin_CeCYP16-like domain of the serologically-defined colon cancer antigen 10 from homo sapiens (PDB ID: 2HQ6), molecular orbital environment (MOE, 2019) software was used. All of the compounds’ structures were drawn using ChemDraw Ultra 12.02, and these structures were saved as MDL files (“.sdf”) for MOE to show. The protein data bank (http://www.rcsb.org/pdb, accessed on 5 May 2023) provided the crystal structures of the colon cancer cell line (2HQ6) and *E. coli* (3T88). Following the removal of the water molecules surrounding the protein, hydrogen atoms were added. Using the MMFF94x force field, the parameters and charges were assigned. Using the DOCK module of MOE, our compounds were docked in the active site following the creation of alpha-site spheres using the site finder module of MOE. The MOE program’s dock scoring was calculated using the London dG scoring formula, placement: triangle matcher, retain 10, and refinement: force field. The leading conformations of the docked ligands were determined by taking into account the RMSD values, binding energies, and binding modes with the chosen residues.

### 2.12. Statistical Analysis

GraphPad Prism (version 5.0, San Francisco, CA, USA) was used to evaluate the results, which were given as means standard deviations of means (SD). Using the Student’s t-test, *p* ≤ 0.05 was judged as significant statistically. Antibacterial activity, as well as antibiotic sensitivity results, were carried out using the R software (version 3.6.1, https://www.R-project.org/, accessed on 15 April 2023).

## 3. Results

### 3.1. Antibacterial Impact of Different Plant Extracts Using Different Solvents

Agar-well diffusion assay was applied to examine the antibacterial action of various extracts of S. officinalis L., *Z. spina-christi* L., and H. sabdariffa L. upon extraction using ethanol, methanol, and water, respectively. Results revealed that, among the examined plant extracts, H. sabdariffa L. extracts were the most effective against all tested bacteria as illustrated in Table 1 and Figure 1. Furthermore, H. sabdariffa L. methanolic extract gave the highest inhibition zone of 39.6 ± 0.20 mm toward E. coli (Figure 1). It could be noticed that *Z. spina-christi* L. water extract has no impact on all tested bacteria, except P. aeruginosa 13.3 ± 0.15 mm. S. officinalis L. ethanol extract displayed the maximum inhibition zone of 27.6 ± 0.3 versus E. coli as compared to ethanol extract of *Z. spina-christi* L.

### 3.2. Measurement of MIC and MBC

Methanol extract of *H. sabdariffa* L. was selected as it had the most promising action. Data presented in Table 2 showed that methanol extract of *H. sabdariffa* L. displayed the lowest MIC and MBC values of 1.9 µg/mL and 3.8 µg/mL, respectively, against *E. cloacae*, followed by *E. coli* that showed MIC and MBC values of 3.99 µg/mL and 7.8 9 µg/mL, respectively. While *S. aureus* recorded the highest MIC and MBC levels of 31.25 µg/mL and 62.5 µg/mL, respectively.

### 3.3. Testing Impact of Antibiotics

The antibiotic sensitivity and resistance pattern for all tested bacterial strains are represented in Table 3. Of the tested bacterial strains, 100% showed resistance to vancomycin (VA), meropenem (MEM), erythromycin (E), amoxicillin/clavulanic acid (AMC), ceftazidime (CAZ), and cefotaxime (CTX) followed by gentamicin (CN), amikacin (AK), and ciprofloxacin (CIP) 75% for each. While 50% of tested bacterial strains were susceptible to tetracycline (TE), doxycycline (DO), and piperacillin/tazobactam (TZP) (Figure 2). Additionally, it was noticed that 50% of examined bacteria were intermediately susceptible to piperacillin/tazobactam (TZP). Therefore, piperacillin/tazobactam (TZP) is the most active agent against tested bacteria.

### 3.4. Testing Synergism among (TZP) and H. sabdariffa L. Extract

Combination of the most promising antibiotic (TZP) with methanol extract of *H. sabdariffa* L. showed synergistic activity toward *E. coli* with a maximum inhibition zone of (63 mm) followed by *E. cloacae* (57mm). While the combination did not show any synergism in the case of *P. aeruginosa* and *S. aureus* as depicted in Table 4.

### 3.5. Scanning Electron Microscopy

Untreated *E. coli* has regularly condensed rods upon investigation using scanning electron microscope, while treating *E. coli* with TZP led to irregularities in the surface and size of cells; however, the application of methanol extract *H. sabdariffa* L. led to the shrinkage of cells and lysis of barriers between cells. Furthermore, the application of a combination of extract and antibiotic (TZP) enhanced the lysis of cells confirming a synergistic effect as depicted in Figure 3.

### 3.6. Anticancer Role of H. sabdariffa L. Extract and Cytotoxicity

*H. sabdariffa* L. methanolic extract was tested versus Caco-2 cells showing promising anticancer activity with IC_50_ of 17.51 ± 0.07 µg/mL. While testing the extract on Vero cells reflected the potency of the extract with CC_50_ of 165.24 ± 0.89 µg/mL as depicted in Figure 4.

### 3.7. Testing the Apoptotic Role of H. sabdariffa L. Extract

Treating Caco-2 cells with the methanol extract of *H. sabdariffa* L. dramatically enhanced (*p* ≤ 0.05) the apoptotic percentage from 10% of untreated cells to 40% of treated cells, confirming the anticancer role of the extract as shown in Figure 4.

### 3.8. GC-MS Analysis of H. sabdariffa L. Methanolic Extract

In order to determine the major bioactive molecules in *H. sabdariffa* L. methanolic extract, GC-MS analysis was performed. The results showed the existence of about 84 different compounds. The major molecules included: n-hexadecanoic acid, hexadecanoic acid, methyl ester, oleic acid, 3-hydroxypropyl ester, cis-13-octadecenoic acid, butanedioic acid,1-hydroxy-2,2-dimethyl-, dimethyl, (R)-11-octadecenoic acid, methyl ester, and 9-octadecenoic acid (Z)-, 2-hydroxy-1-(hydroxymethyl) ethyl ester, as illustrated in Table 5, Figure 5.

### 3.9. Molecular Modeling: Docking Study

The molecular docking studies have been performed on n-hexadecanoic acid, hexadecanoic acid-methyl ester, and oleic acid,3-hydroxypropyl ester against the crystal structure of *E. coli* (3T88) and a colon cancer cell line (2HQ6) as further evidence for biological screening investigations. The results indicated good agreement between the docking and experimental results. Our predictions based on the docking showed that the inhibitor compounds will interact effectively with specific proteins coupled to their active sites. The RMSD values for all compounds seemed acceptable. Data represented in Table 6 revealed that the crystal structure of *E. coli* (3T88) showed good docking scores and binding interactions with the tested compounds, which were observed (−6.86665 kcal mol^−1^, −6.10658 kcal mol^−1^, and −7.58806 kcal mol^−1^), respectively. We noticed that n-hexadecanoic acid interacts with *E. coli* (3T88) receptors via ASN 202 through its O 49 atom. Additionally, hexadecanoic acid-methyl ester interacts with *E. coli* (3T88) by donating H atoms of O 48 and O 50 through CYS 240 and ASP 239 amino acids receptors, in addition to accepting H atom between O 48 of the molecule and ARG 173 amino acid residue. On the other side, the interaction of oleic acid,3-hydroxypropyl ester with *E. coli* (3T88) formed two hydrogen bonds (donating and accepting) through O 19 in the compound via CYS 141 and THR 144 amino acids in the receptor (Table 7). Similarly, the docking score values for examined compounds with the colon cancer cell line (2HQ6) protein are (−7.06866 kcal mol^−1^, −6.90753 kcal mol^−1^, and −7.36223 kcal mol^−1^), respectively. A colon cancer cell line (2HQ6) protein interacts with n-hexadecanoic acid via the residues (ALA 102, SER 111, and GLN 112), while results in the interaction with hexadecanoic acid-methyl ester through O 48 by accepting the H atom between GLY 110 active site residue of the colon cancer cell line (2HQ6) protein. Oleic acid,3-hydroxypropyl ester has a strong interaction with the active site bound to chain (A) of the colon cancer cell line (2HQ6) by amino acid pocket molecules (GLN 112) (Table 8 and Table 9). Figure 6 and Figure 7 illustrate the compounds’ effective docking poses for *E. coli* (3T88) and the colon cancer cell line (2HQ6), and in vitro inhibitory activities are illustrated by increased interaction between substances and receptors.

## 4. Discussion

Overuse of antibiotics is a developing global issue that has an impact on the economy, the ecosystem, humans, and livestock. One of the biggest risks to healthcare system is the possibility that many significant clinical bacteria are becoming resistant to traditional antibiotics. The issue is made worse by the unavailability of new medications since they are expensive and time-consuming to develop. Thus, finding new alternative antimicrobial medicines is essential for treating microbial infections. Plant extracts have significant components of a novel approach to battle pathogenic bacteria since they provide abundant resources for bioactive chemicals [42].

Medicinal plants have historically been employed as medications by various cultures for ages because of their preventive and curative qualities [43]. According to the solvent’s polarization, synthetic organic solvents have a strong ability to extract both hydrophilic and hydrophobic molecules of plants. Most solvents are volatile chemical compounds that are employed to separate functional biomolecules [44].

In the current investigation, ethanol, methanol, and water extracts of *S. officinalis* L., *Z. spina-christi* L., and *H. sabdariffa* L. showed various antibacterial actions against *E. cloacae* (ATCC13047), *S. aureus* (ATCC 25923), *P. aeruginosa* (RCMB008001), and *E. coli* (RCMB004001) where *H. sabdariffa* L. methanolic extract had the highest inhibition zone toward tested bacteria, especially *E. coli*. In accordance with Abdallah, [24] who reported the antibacterial role of *H. sabdariffa* L. versus *Acinetobacter baumannii*. Furthermore, Márquez-Rodríguez [45] illustrated the beneficial role of *H. sabdariffa* L. extract toward foodborne pathogens.

The present findings showed the MIC and MBC of methanol extract from *H. sabdariffa* toward various bacterial strains. Whenever the MBC/MIC ratio is less than 4, the extract is thought to be bactericidal, and when it is greater than 4, it is thought to be bacteriostatic [46]. Thus, *H. sabdariffa* has a bactericidal action versus most of the tested bacteria. There has been evidence of antibacterial action against MDR clinical strains in prior research on herbal remedies [47,48].

The incremental transformations in the structure of *E. coli* was evaluated in this work using scanning electron microscopy after multiple treatments to the cell regions that the methanol extract and antibiotics target, causing cell division and shrinkage in accordance with several studies [49,50,51], which reported distortion in the surface of bacterial stains upon treatment using essential oils of plant extracts.

In the current work, methanol extract of *H. sabdariffa* L. exhibited promising anticancer action versus human colonic cancer cells (Caco-2) through an increasing apoptotic percentage of cells and minimal cytotoxicity versus normal cells to be applied in the drug industry. This is in accordance with Malacrida et al. [52], who reported the progressive role of *H. sabdariffa* toward human breast cancer cells. Additionally, Xu et al. [53] explained the mode of action of *H. sabdariffa* to kill colorectal cancer cells. An analysis of *H. sabdariffa* revealed the presence of different fatty acids and their derivatives, which contribute to its antibacterial activity as well as synergistic action with tested antibiotics. It was reported that OH groups of fatty acids affected the cell membrane of bacteria [54]. Because of the amphipathic character of fatty acids, they can solubilize a variety of membrane components, including proteins and lipid bilayers that may lead to cell lysis [55]. This is in agreement with Hesham et al. [56], who stated the presence of fifteen compounds with major lipid content with various biological activities of *H. sabdariffa*. Additionally, Vijayakumar et al. [57] reported the presence of seven phytoconstituents, such as 1, 2-benzenedicarboxylic acid, octadecanoic acid, 3-n-hexylthiolane, 1-iodoundecane, 2, 2, 4—trimethyl 3-pentanone, 2-propenamide, and amyl nitrite in the methanolic flower extracts of *H. rosa Sinensis* on the GC-MS analysis. Additionally, Mohammed [58] reported the synergistic action of some antibiotics with *Hibiscus sabdariffa* extract versus *Salmonella typhi*. A sort of bioinformatics modeling known as molecular docking involves the combination of two or more molecules to generate a stable adduct. Molecular docking generates different possible adduct structures that are ranked and grouped together using a scoring function in the software. The information obtained from the docking technique can be used to suggest the binding energy, free energy, and stability of the ligand [59]. In the present work, the biological screening investigations have been supported by molecular docking studies. It is becoming increasingly significant in the process of rational medication design. It is well-known that precise pharmacological activity is caused by the chemical binding of one molecule (the ligand) to the pocket of another molecule (the receptor), which is frequently a protein. When the structure of proteins is known, protein-ligand docking refers to the search for the precise ligand conformations within a specific protein. In the current study, the major compounds obtained from *H. sabdariffa* L. methanolic extract exhibited good interaction with the crystal structure of *E. coli* MenB: OSB-NCoA (PDB ID:3T88) and structure of the cyclophilin -CeCYP16(PDB ID:2HQ6) of the colon cancer that were recovered from the protein data bank. The analysis of these docked ligands with these proteins displayed inhibition against *E. coil* and colon cancer. It was reported that humans lack the menaquinone (vitamin K2) production route. While in *E. coli*, menaquinone production is extensively investigated [60]. Men A, B, D, and E are confirmed as prospective targets for structure-based antibacterial drug development among the enzymes MenA-F participating in this system [61]. Menaquinone functions as an electron transporter in the respiratory chain during the anaerobic development of *E coli* [60]. In *E coli*, *Bacillus subtilis*, and *Mycobacterium phlei*, the production of menaquinone from chorismate was initially investigated. The enzyme 1,4-dihydroxynaphthoyl-CoA synthase (MenB), which catalyzes the synthesis of the second aromatic ring of naphthoquinone through a Claisen (or Dieckmann) condensation is involved in one significant step in this pathway. The succinyl side chain of O-succinylbenzoate (OSB) undergoes this condensation. As demonstrated in reactions catalyzed by -ketoacyl-ACP synthases, in typical Claisen condensations, both the nucleophile and electrophile are activated by thioester production. The MenB reaction, by contrast, is distinctive because it exclusively activates the nucleophile [62].

It was investigated that when 1,4-dihydroxynaphthoyl-CoA synthase (MenB) is inhibited as a drug target, it would disrupt the biosynthesis of menaquinone, which is essential for the electron transport chain in certain bacteria. In bacteria, such as *E. coli*, menaquinone is essential for the generation of energy through the anaerobic respiration process. Additionally, MenB inhibition would result in the deficiency of menaquinone, which would affect the capacity of bacteria to produce ATP and engage in anaerobic respiration. This disruption in the respiratory chain may negatively impact bacterial growth and survival as our results showed, making it a potential strategy for antimicrobial therapy. In line with our results, Matarlo et al. [62] showed that the synthesized compound 4-oxo-4-phenyl-2-butenoyl methyl esters undergo a reaction to form the corresponding CoA adduct upon entering the cell. As a result, this adduct inhibits the enzyme MenB, leading to a reduction in the levels of DHNA (1,4-dihydroxy-2-naphthoic acid) and MK (menaquinone) and finally inhibiting the growth of *Staphylococcus aureus*. Furthermore, Li et al. [61] found a group of 1,4-benzoxazines that, in laboratory experiments, showed potential antibacterial action against mycobacterium TB H37Rv, which is consistent with our findings.

Regarding cyclophilins, they are a group of enzymes that catalyze the cis-trans isomerization of peptide bonds involving proline residues in proteins. They are also known as peptidyl-prolyl isomerases (PPIases). They are essential for the maturation, folding, and control of protein function. Due to their involvement in numerous disease processes and their capacity to modify protein function, PPIases are significant as therapeutic targets. For instance, their role in protein folding, where they help ensure the proper folding of newly synthesized proteins [63], functions in regulating signaling pathways, where they can interact with signaling proteins and control their activity by isomerizing proline-containing motifs [64], and viral infections, where they have been discovered to play a role in the replication and spread of specific viruses [65], It is important to note that PPIase-targeted medication development is still an active field of study and that it can be difficult to target PPIases in particular without also impacting other crucial cellular functions.

It was emphasized that protein misfolding or improper folding as a result of PPIase inhibition can cause protein aggregation, malfunction, and loss of protein activity. A change in cellular responses could result from inhibiting PPIases since it can modify these connections and interfere with downstream signaling pathways. This disruption can be therapeutically advantageous in diseases where abnormal signaling pathways contribute to the pathogenesis, such as cancer. Inhibiting PPIases can disrupt the viral life cycle, inhibiting viral replication and reducing viral load. This makes PPIases attractive targets for antiviral drug development, as inhibiting their activity can help combat viral infections. Targeting this enzyme, which is discovered to be more abundant in cancer cells, has the potential to serve as a viable strategy for developing a supplementary treatment that can improve the effectiveness of anticancer therapies. In the same line, Uchida et al. [66] reported that PPIase–Parvulin inhibitor (PiB) was one of the most effective inhibitors of the proliferation of numerous cancer lines. The recently identified prolyl isomerase (Pin1) inhibitor, KPT-6566, has been shown to inhibit the enzymatic activity of PPIase and induce its degradation by creating a covalent binding with its catalytic core [67]. Furthermore, it has been demonstrated that inhibiting Pin1 increases the sensitivity of certain cancer cells to chemotherapeutic treatments. For example, when Pin1 is inhibited, hepatocellular carcinoma (HCC) cells have been shown to become sensitive to sorafenib [68], breast cancer cells to trastuzumab [69] and rapamycin [70], and colon cancer cells to taxol [71]. Overall, these data indicated that PPIase inhibition is a promising target for cancer therapy

## 5. Conclusions

The current work evaluated the antibacterial capability of various medicinal plants using different solvents. Results showed that *H. sabdariffa* L. methanolic extract displayed the highest antibacterial activity against all tested bacteria. In addition, the methanol extract of *H. sabdariffa* L. showed promising anticancer potential against human colonic cancer cells (Caco-2). GC-MS analysis revealed the presence of bioactive metabolites with various biological functions, which are fatty acids and their derivatives. The promising antibacterial and anticancer activities were confirmed by in silico studies of the most commonly identified bioactive compounds from *H. sabdariffa* L. methanolic extract. Thus, *H. sabdariffa* L. can be used as a natural alternative agent for biomedical applications.

## Data Availability

The data presented in this study are available on request from the corresponding author.

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
