# Peer review of "GC-MS Analysis, Antibacterial, and Anticancer Activities of Hibiscus sabdariffa L. Methanolic Extract: In Vitro and In Silico Studies"

_microorganisms, 2023, doi:10.3390/microorganisms11061601_

Round 1
Reviewer 1 Report
The current work evaluated the antibacterial capability of various medicinal plants using different solvents.
Results showed that H. sabdariffa L. methanolic extract displayed the highest antibacterial activity against all tested bacteria.
In addition, the methanol extract of H. sabdariffa L. showed promising anticancer potential against Human colonic cancer cells (Caco-2).
GC-MS analysis revealed the presence of bioactive metabolites with various biological functions which are fatty acids and their derivatives.
The promising antibacterial and anticancer activities were confirmed by in silico studies of the most commonly identified bioactive compounds from H. sabdariffa L. methanolic extract.
Thus, H. sabdariffa L. can be used as a natural alternative agent for biomedical applications.

Author Response
We highly appreciate reviewer valuable comments and through review for our manuscript.
Regarding the manuscript sections: reviewer support our approach and findings and we highly appreciate reviewer valuable comments.
We hope our manuscript now is acceptable in the current form for publication.
Reviewer 2 Report
A Review on “ GC-MS analysis, antibacterial, and anticancer activities of Hibiscus 2 sabdariffa L. methanolic extract: In vitro and in silico studies” a well written manuscript. The authors have done the antibacterial and anticancerous activity of Hibiscus sabdariffa extract . The author also performed in-silico and cell line studies.
Major comments:
· Manuscript must be formatted correctly, a lot of formatting issues observed, headings, fonts, spaces, italics must be observed and corrected
· Since flower of hibiscus was used for extraction, its is obvious that it contains a lot of flavonols, terpenes, and phenolic compounds etc. It is strange that author has identified mostly -oic acids. Does the author find the chromatogram strange? Did the author not expect to identify the high antioxidant content in the flower? Support with suitable literature
· Why did author refrain to use name of the receptors used for the study, what do you mean by crystal Structure of E. coli (3T88) 190 and Colon cancer cell line (2HQ6) Line 190
· Since the top lead compounds have long alkyl chains and of course only functions groups are mean for bond formation, in that case the alkyl chain might have a lot of flexibility in the binding cavity. It is suggested that author should perform 100 ns of MDS to observe the changes.
· How did author validate the docking program?
· What was the criteria for selecting receptors? Targeting multiple potential targets for both antibacterial and anticancer could have been give more detailed insights about the activity.
Minor comments:
1. The heading font needs to be corrected.
2. Remove the extra bracket in Ln 20
3. The title needs to be revised. The activity for three plants has been checked but title has only one plant name.
4. LN 26-27 is not clear.
5. Name of the organisms needs to be itlaicised throughout.
6. Ln 38, the PDB id is for which receptor protein of E. coli. What is being targeted?
7. Keyword antitumor is written in what context. Study is about antibacterial and anticancerous activity.
8. Introduction LN83 “previous” P needs to be capitalised.; Ln 89 Is the study is about anticancerous or antitumor. Authors need to finalise one effect and accordingly do the corrections in the entire manuscript.
9. Entire material and methods section has been italicised. Needs to be corrected.
10. References mentioned in material and methods are for protocol purpose? If yes please make in-line citations and not at the end of each section.
11. What was the number of ligands used for docking purpose?
12. The protein has 6 chains in its crystal structure. Was the entire protein used for docking or only chain?
13. Results Ln 213 is not clear.
14. What does repetition in table 6 indicates?
Minor english corrections: acronym, spacing, italic, grammer and spelling must be checked correctly
Author Response
We highly appreciate the reviewers’ insightful and helpful comments on our manuscript.
We have followed all the instructions and all the required formats have been done. Below is our point-by-point response to each comment.
Major comments:
Manuscript must be formatted correctly, a lot of formatting issues observed, headings, fonts, spaces, italics must be observed and corrected.
- Thanks a lot for your comment. All mistakes in the manuscript have been revised and corrected which have been highlighted in the revised manuscript
Since flower of hibiscus was used for extraction, its is obvious that it contains a lot of flavonols, terpenes, and phenolic compounds etc. It is strange that author has identified mostly -oic acids. Does the author find the chromatogram strange? Did the author not expect to identify the high antioxidant content in the flower? Support with suitable literature .
- We highly appreciate your helpful comments, and we totally agree with you. The results of GC-MS analysis revealed the presence of 84 different bioactive metabolites in the methanol extract of Hibiscus but based on the concentration (peak area %) the most predominant compounds were fatty acids and their derivatives ester including n -hexadecanic acid 6% followed by hexadecanic acid methyl ester 3% and oleic acid 2.27%.it was reported that these compounds had different biological activities. So, the antibacterial and anticancer activities of hibiscus extract may be contributed to the presence of these compounds. Thus, these compounds were chosen for docking assay to study and confirm their biological activities.
Vijayakumar et al. (2018) reported the presence of different phytoconstituents such as 1, 2-Benzenedicarboxylic acid, Octadecanoic acid, 3- N-Hexylthiolane, 1-Iodoundecane, 2, 2, 4 - Trimethyl 3-pentanone, 2-Propenamide and Amyl nitrite in the methanolic flower extracts of H. rosa Sinensis up on the GC-MS analysis, in agreement with our results.
Why did author refrain to use name of the receptors used for the study, what do you mean by crystal Structure of E. coli (3T88) 190 and Colon cancer cell line (2HQ6) Line 190
- (3T88) and (2HQ6) due to protein data bank codes of coli and colon cancer crystal structure respectively.
Since the top lead compounds have long alkyl chains and of course only functions groups are mean for bond formation, in that case the alkyl chain might have a lot of flexibility in the binding cavity. It is suggested that author should perform 100 ns of MDS to observe the changes.
- Although the docking simulation procedure took a variety of times to cover all tested compounds, the ligands were inserted in the site using the triangle matcher approach after the general docking scenario was run for 100 ns on the stiff receptor atoms. The GBVI/WSA dG procedures for rescoring were used, along with the London dG as a scoring function. The best five conformations that were present in the crystal structure and had a lower RMSD value were predicted by the docking process.
How did author validate the docking program?
- By scoring energy values, the binding rank of the investigated substances was distinguished. For docked compounds, additional features included ligand type, receptors, interaction type, H-bond length, and energy content. The determining factor for the extent of interaction validity overall was expanded to include the H-bond length (<3.5).
What was the criteria for selecting receptors? Targeting multiple potential targets for both antibacterial and anticancer could have been give more detailed insights about the activity.
- On the basis of our investigation, the compounds which were theoretically examined here exhibited good docking scores and binding interactions. Therefore, this molecule may be an effective therapeutic candidate, and their effectiveness against coli (3T88) and colon cancer cell line (2HQ6) more strongly than other types of proteins.
Minor comments:
- The heading font needs to be corrected.
Thanks for your comment. The heading has been corrected
- GC-MS analysis, antibacterial, and anticancer activities of Hibiscus sabdariffa methanolic extract: In vitro and in silico studies
- Remove the extra bracket in Ln 20
- Much thanks. It has been removed
- The title needs to be revised. The activity for three plants has been checked but title has only one plant name.
- We highly appreciate your helpful comment. Concerning the title, three plant extracts have been screened for their antibacterial activity. Out of them, only the methanol extract of sabdariffa L. was the most effective against all tested bacteria. Thus, it has been chosen for further study as anticancer activity, GC-MS analysis, and docking study. so, the title is focused only on the Hibiscus plant.
- LN 26-27 is not
Thanks a lot for your comment. The result has been cleared as below.
While 50 % of tested bacteria were sensitive and 50 % were intermediate sensitive to piperacillin/tazobactam (TZP) based on the inhibition zone but still less than the extract.
- Name of the organisms needs to be italicized throughout.
- Much thanks for your kind comment. The name of the organisms has been italicized in the manuscript
- Ln 38, the PDB id is for which receptor protein of coli. What is being targeted?
- (3T88) the receptor protein of coli that has been targeted and downloaded from Protein Data Bank PDB (http://www.rcsb.org/pdb).
- Keyword antitumor is written in what context. Study is about antibacterial and anticancerous activity.
Thanks a lot. It has been corrected.
- Keywords: Medicinal plants; antibacterial; MDR; anticancer; GC-MC; Scanning electron microscope; docking study
- Introduction LN83 “previous” P needs to be capitalised.; Ln 89 Is the study is about anticancerous or antitumor. Authors need to finalise one effect and accordingly do the corrections in the entire manuscript.
- We highly appreciate your helpful comment. We have followed all the instructions.
- Entire material and methods section has been italicised. Needs to be corrected.
- Much thanks for your kind comment. It has been corrected and highlighted in the manuscript.
- References mentioned in material and methods are for protocol purpose? If yes please make in-line citations and not at the end of each section
- We totally agree with this suggestion. It has been corrected as required.
- . Materials and Methods
- 1. Plant materials and extraction
- Two plants were bought in dried form from an Egyptian local market: S. officinalis L. leaves, often known as sage, and H. sabdariffa L. flowers, also known as roselle. Additionally, the professional plant-qualified worker at the Botany and Microbiology Department, Faculty of Science, Benha University confirmed Z. spina-christi L. leaves that had been taken from Monufia 30.52°N 30.99°E in Egypt (Voucher No. ID: 0045).
- Sage, Ziziphus, and Rosella powder samples weighing 10 g each were used in the experiment. For water extraction, the plant materials were heated at 90°C for 30 minutes in 90 mL of distilled water, then overnight at 37°C and 150 rpm in a shaking incubator. Similar to this, 10 g of each examined plant material powder was combined with ethanol and methanol (99%) separately, before incubation at 37°C and 150 rpm for an entire night. The resultant liquid extracts were then passed through a Whatman No. 1 filter to separate them from the solid residue and then were evaporated to room temperature [25-27]. For testing their antibacterial activity, aqueous extracts were dissolved in distilled water, while methanol and ethanol extracts were diluted in 10% Dimethyl Sulfoxide (DMSO) to achieve final concentrations of 10mg/mL .
- 2. Bacterial strains
- Pathogenic bacterial strains were provided by AL-Azhar University, Regional Center for Mycology and Biotechnology (RCMB) culture collection unit. The bacteria used in this study were E. cloacae (ATCC13047), S. aureus (ATCC 25923), P. aeruginosa (RCMB008001) and E. coli (RCMB004001).
- 3. Antibacterial assay
- The tested bacteria were grown in nutrient broth and cultured overnight at 37∘C to achieve the turbidity of 0.5McFarland standards giving 1.5 × 108 CFU/mL. Antibacterial activity of plant extracts was performed using agar well diffusion method [28, 29]. Mueller Hinton agar plates were cultured with bacterial suspensions. Wells (6 mm) were drilled into the inoculated media using a sterile cork- borer. 100 µL of plant extracts (aqueous, methanol, and ethanol) were poured separately into each well. Then, the plates were kept for 30 in the refrigerator to better diffusion of the plant extracts into the agar. Penicillin G (P, 10 mg) and DMSO (10 %) were used as positive and negative controls. The plates were incubated at 37oC for 24 h. After the incubation period, the zone of inhibition was measured to determine the antibacterial capability.
- 4. Determination of minimum inhibitory concentration (MIC) and minimum bactericidal concentration (MBC)
- For MIC detection: The plant extract that displayed the highest bacterial growth inhibition using agar well diffusion technique was further screened by micro-dilution technique [30, 31]. Using broth as diluent, two-fold serial dilutions of methanol extract were prepared from the stock solution to achieve concentrations ranging from (1000 to 1.9 µg /mL). Finally, a volume of 10 μL was taken from bacterial suspension (105 CFU/mL) and then added to each well. To ensure the sterility and purity of the medium, wells containing un-inoculated medium with and without extract were used as a control. In order to ensure that the organism could grow in the medium, a third control well was used that contained inoculated medium but no extract. The turbidity was tested as a measure of microbial growth after being incubated for 24 h at 37 °C. The lowest concentration of plant extract dilution which inhibited any visible growth of the tested bacteria is considered MIC value.
- For MBC detection: On Mueller Hinton Agar plates, streaks from the plant extract's lowest concentrations that had no discernible growth were spread out. Then, the plates were incubated at 37 degrees Celsius for 24 h before being checked for bacterial growth in accordance with plant extract concentration. MBC was found to be the lowest concentration of plant extract on the newly inoculated agar plates that did not show any bacterial growth. Two types of activities were demonstrated by plant extract: a bacteriostatic (MBC/MIC ≥4) and bactericidal (MBC/MIC <4) [32].
- What was the number of ligands used for docking purpose?
- Three ligands used for docking (n-Hexadecanoic acid, Hexadecanoic acid-methyl ester, and Oleic acid,3-hydroxypropyl ester)
- The protein has 6 chains in its crystal structure. Was the entire protein used for docking or only chain?
- Only the primary chain was docked. The main chain has been kept firm, While the side chains are still flexible. This approximation allows the side chains of the proteins to find the position in which the interactions are most favorable.
- Results Ln 213 is not clear.
Thanks a lot, it has been cleared as below
Furthermore, H. sabdariffa L. methanolic extract gave the highest inhibition zone of 39.6 ± 0.20 mm towards E. coli
- What does repetition in table 6 indicates?
- The repetition in table 6 due to the best five conformations of ligand that were present in the crystal structure and had a lower RMSD.
We hope our manuscript now is acceptable in the current form for publication.

Round 2
Reviewer 2 Report
Kindly mention the name of the receptors used for docking, just the PDB IDs are not enough. Also, highlight their importance as the drug target. Support with relevant literature, why these drug targets are essential? What will happen when you inhibit these targets? who else has targeted these receptors in their studies and how relevant are your findings compared to them?
Kindly run the MD simulation of receptor-ligand complex to study the interaction patterns with binding site residues. I again reiterate that your ligands have only functional groups acting as electron donors or acceptors, the rest of the molecule is just a long aliphatic chain. Hence I recommend the author to conduct MDS at least for 100 ns and evaluate the findings.
Done already by authors, however, once careful observation would further improve
Author Response
We highly appreciate the reviewers’ insightful and helpful comments on our manuscript
Kindly mention the name of the receptors used for docking, just the PDB IDs are not enough
Thanks a lot for your kind comment. The name of the receptors:
Crystal structure of Escherichia coli MenB in complex with substrate analogue, OSB-NCoA (PDB ID:3T88) and Structure of the Cyclophilin_CeCYP16-Like Domain of the Serologically Defined Colon Cancer Antigen 10 from Homo Sapiens (PDB ID: 2HQ6). The co-ligand was removed from both proteins in order to study the binding effect of our examined ligands.
highlight their importance as the drug target
1- Crystal structure of Escherichia coli MenB in complex with substrate analogue, OSB-NCoA (3T88):
Menaquinone (vitamin K2) biosynthesis pathway is absent in humans. Menaquinone biosynthesis is extensively studied in E. coli (1). Among the enzymes MenA–F involved in this pathway, Men A, B, D and E are validated as potential targets for structure-based anti-bacterial drug discovery (2). During anaerobic growth of Escherichia coli, menaquinone serves as an electron carrier in the respiratory chain (1). The synthesis of menaquinone from chorismate was initially studied in Escherichia coli, Bacillus subtilis, and Mycobacterium phlei. One important step in this pathway involves the enzyme 1,4-dihydroxynaphthoyl-CoA synthase (MenB), which catalyzes the formation of the second aromatic ring of naphthoquinone through a Claisen (or Dieckmann) condensation. This condensation occurs within the succinyl side chain of O-succinylbenzoate (OSB). In typical Claisen condensations, both the nucleophile and electrophile are activated through thioester formation, as seen in reactions catalyzed by β-ketoacyl-ACP synthases. However, the MenB reaction is unique because it only activates the nucleophile (4).
What will happen when you inhibit these targets?
- When 1,4-dihydroxynaphthoyl-CoA synthase (MenB) is inhibited as a drug target, it would disrupt the biosynthesis of menaquinone, which is essential for the electron transport chain in certain bacteria. Menaquinone plays a crucial role in the generation of energy through anaerobic respiration in bacteria like Escherichia coli. Also, inhibiting MenB would lead to a deficiency of menaquinone, which could impair the bacteria's ability to generate ATP efficiently and perform anaerobic respiration. This disruption in the respiratory chain can have detrimental effects on bacterial growth and survival as our resulted showed, making it a potential strategy for antimicrobial therapy.
who else has targeted these receptors in their studies and how relevant are your findings compared to them?
- In agreement with our results Li et al. (2), identified a set of 1,4-benzoxazines, showing potential antibacterial activity against Mycobacterium tuberculosis H37Rv in laboratory tests. These compounds were identified through screening a compound library against the enzyme 1,4-dihydroxy-2-naphthoyl-CoA (DHNA-CoA) synthase (MenB) in the menaquinone biosynthesis pathway of tuberculosis. Also, Matarlo et al. (4) showed that the synthesized compound 4-oxo-4-phenyl-2-butenoyl methyl esters, upon entering the cell, undergoes a reaction to form the corresponding CoA adduct. This adduct inhibits the enzyme MenB, resulting in a reduction in the levels of DHNA (1,4-dihydroxy-2-naphthoic acid) and MK (menaquinone) and finally the growth inhibition of Staphylococcus aureus.
- Jiang, M., Cao, Y., Guo, Z. F., Chen, M., Chen, X., & Guo, Z. (2007). Menaquinone biosynthesis in Escherichia coli: identification of 2-succinyl-5-enolpyruvyl-6-hydroxy-3-cyclohexene-1-carboxylate as a novel intermediate and re-evaluation of MenD activity. Biochemistry, 46(38), 10979-10989.
- Li, X., Liu, N., Zhang, H., Knudson, S. E., Slayden, R. A., & Tonge, P. J. (2010). Synthesis and SAR studies of 1, 4-benzoxazine MenB inhibitors: Novel antibacterial agents against Mycobacterium tuberculosis. Bioorganic & medicinal chemistry letters, 20(21), 6306-6309.
- Li, H. J., Li, X., Liu, N., Zhang, H., Truglio, J. J., Mishra, S., ... & Tonge, P. J. (2011). Mechanism of the intramolecular Claisen condensation reaction catalyzed by MenB, a crotonase superfamily member. Biochemistry, 50(44), 9532-9544.
- Matarlo, J. S., Lu, Y., Daryaee, F., Daryaee, T., Ruzsicska, B., Walker, S. G., & Tonge, P. J. (2016). A methyl 4-oxo-4-phenylbut-2-enoate with in vivo activity against MRSA that inhibits MenB in the bacterial menaquinone biosynthesis pathway. ACS infectious diseases, 2(5), 329-340.
2-Regarding Structure of the Cyclophilin_CeCYP16-Like Domain of the Serologically Defined Colon Cancer Antigen 10 from Homo Sapiens (2HQ6):
- Cyclophilins (peptidyl-prolyl isomerases (PPIases)) are a class of enzymes that catalyze the cis-trans isomerization of peptide bonds involving proline residues in proteins. They play a crucial role in protein folding, maturation, and regulation of protein function. The importance of PPIases as drug targets lies in their involvement in various disease processes and their potential to modulate protein function. For example, their role in protein folding, it assists in the correct folding of newly synthesized proteins (1), modulation of signaling pathways, it can interact with signaling proteins and regulate their activity by isomerizing proline-containing motifs (2), and viral infections it have been found to play a role in the replication and propagation of certain viruses (3). However, it's worth noting that the development of PPIase-targeted drugs is still an active area of research, and there are challenges associated with specifically targeting PPIases without affecting other essential cellular processes.
What will happen when you inhibit these targets?
- PPIases play a crucial role in protein folding, assisting in the proper conformational changes required for functional proteins. Inhibiting PPIases can lead to misfolding or improper folding of proteins, which can result in protein aggregation, dysfunction, and loss of protein activity. Inhibiting PPIases can disrupt these interactions and interfere with downstream signaling pathways, potentially leading to altered cellular responses. This disruption can be therapeutically advantageous in diseases where abnormal signaling pathways contribute to the pathogenesis, such as cancer. Inhibiting PPIases can disrupt the viral life cycle, inhibiting viral replication and reducing viral load. This makes PPIases attractive targets for antiviral drug development, as inhibiting their activity can help combat viral infections. Modulation of immune responses: PPIases are involved in regulating immune responses by modulating immune signaling molecules and transcription factors. Inhibiting PPIases can alter immune cell differentiation, cytokine production, and immune system activity. This modulation can be beneficial in conditions where immune dysregulation occurs, such as autoimmune diseases or transplant rejection, where reducing immune activity may be desired.
who else has targeted these receptors in their studies and how relevant are your findings compared to them?
- Targeting this enzyme, which is discovered to be more abundant in cancer cells, has the potential to serve as a viable strategy for developing a supplementary treatment that can improve the effectiveness of anticancer therapies. In the same line, Uchida et al. (4), found that PPIase-Parvulin Inhibitor (PiB) was one of the most potent inhibitors of proliferation of several cancer lines. Juglone, a compound derived from walnut trees, and its derivative buparvaquone have been identified as Pin1 inhibitors. They achieve this by covalently modifying the catalytic core of the Pin1 enzyme (5). Juglone has demonstrated effectiveness in suppressing various types of cancer cells and is commonly used in Pin1 research. Additionally, cis-locked alkene peptidomimetics act as Pin1 inhibitors by mimicking Pin1's substrate, leading to antiproliferative effects in ovarian cancer cells (6). Imazamethabenz, an herbicide containing an imidazoline ketone, inhibits migration and invasion while promoting apoptosis in breast cancer cells by directly interacting with Pin1 (7). A newly discovered Pin1 inhibitor, KPT-6566, has been found to inhibit the enzymatic activity of PPIase and induce its degradation by forming a covalent bond with its catalytic core (8). In addition, blocking Pin1 has been shown to enhance the sensitivity of various cancer cells to chemotherapy drugs. For example, inhibiting Pin1 has been found to sensitize hepatocellular carcinoma (HCC) cells to sorafenib (9), breast cancer cells to trastuzumab (10), and rapamycin (11), and colon cancer cells to Taxol (12). All these evidences suggests that PPIase inhibition is a promising target for cancer therapy.
- Gething, M. J., & Sambrook, J. (1992). Protein folding in the cell. Nature, 355, 33-45.
- Leung, A. W., & Halestrap, A. P. (2008). Recent progress in elucidating the molecular mechanism of the mitochondrial permeability transition pore. Biochimica et Biophysica Acta (BBA)-Bioenergetics, 1777(7-8), 946-952..
- Dorfman, T., Weimann, A., Borsetti, A., Walsh, C. T., & Göttlinger, H. G. (1997). Active-site residues of cyclophilin A are crucial for its incorporation into human immunodeficiency virus type 1 virions. Journal of virology, 71(9), 7110-7113.
- Uchida, T., Takamiya, M., Takahashi, M., Miyashita, H., Ikeda, H., Terada, T., ... & Hunter, T. (2003). Pin1 and Par14 peptidyl prolyl isomerase inhibitors block cell proliferation. Chemistry & biology, 10(1), 15-24.
- Hennig, L., Christner, C., Kipping, M., Schelbert, B., Rücknagel, K. P., Grabley, S., ... & Fischer, G. (1998). Selective inactivation of parvulin-like peptidyl-prolyl cis/trans isomerases by juglone. Biochemistry, 37(17), 5953-5960.
- Wang, X. J., Xu, B., Mullins, A. B., Neiler, F. K., & Etzkorn, F. A. (2004). Conformationally locked isostere of phosphoSer− cis-pro inhibits pin1 23-fold better than phosphoSer− trans-pro isostere. Journal of the American Chemical Society, 126(47), 15533-15542.
- Liu, C., Mu, C., Li, Z., & Xu, L. (2017). Imazamethabenz inhibits human breast cancer cell proliferation, migration and invasion via combination with Pin1. Molecular Medicine Reports, 15(5), 3210-3214.
- Campaner, E., Rustighi, A., Zannini, A., Cristiani, A., Piazza, S., Ciani, Y., ... & Del Sal, G. (2017). A covalent PIN1 inhibitor selectively targets cancer cells by a dual mechanism of action. Nature communications, 8(1), 15772.
- Zheng, M., Xu, H., Liao, X. H., Chen, C. P., Zhang, A. L., Lu, W., ... & Lu, K. P. (2017). Inhibition of the prolyl isomerase Pin1 enhances the ability of sorafenib to induce cell death and inhibit tumor growth in hepatocellular carcinoma. Oncotarget, 8(18), 29771.
- Sajadimajd, S., & Yazdanparast, R. (2017). Sensitizing effect of juglone is mediated by down regulation of Notch1 signaling pathway in trastuzumab-resistant SKBR3 cells. Apoptosis, 22, 135-144.
- Stanya, K. J., Liu, Y., Means, A. R., & Kao, H. Y. (2008). Cdk2 and Pin1 negatively regulate the transcriptional corepressor SMRT. The Journal of cell biology, 183(1), 49-61.
- Min, S. H., Lau, A. W., Lee, T. H., Inuzuka, H., Wei, S., Huang, P., ... & Lu, K. P. (2012). Negative regulation of the stability and tumor suppressor function of Fbw7 by the Pin1 prolyl isomerase. Molecular cell, 46(6), 771-783.
Kindly run the MD simulation of the receptor-ligand complex to study the interaction patterns with binding site residues.
- Thank you for your suggestion to conduct an MD simulation to investigate the interaction patterns of the receptor-ligand complex with the binding site residues. While we appreciate the potential insights that could be gained from such an analysis, we regret to inform you that we currently do not have the necessary resources to perform MD simulations
